# A Novel Peptide from *Polypedates megacephalus* Promotes Wound Healing in Mice

**DOI:** 10.3390/toxins14110753

**Published:** 2022-11-02

**Authors:** Siqi Fu, Canwei Du, Qijian Zhang, Jiayu Liu, Xushuang Zhang, Meichun Deng

**Affiliations:** 1Hunan Key Laboratory of Medical Epigenomics, Department of Dermatology, The Second Xiangya Hospital, Central South University, Changsha 410013, China; 2Chengdu Pep Biomedical Co., Ltd., Chengdu 610041, China; 3Wound Center of Xiangya Hospital, Central South University, Changsha 410013, China; 4Hunan Province Key Laboratory of Basic and Applied Hematology, Department of Biochemistry and Molecular Biology, School of Life Sciences, Central South University, Changsha 410013, China; 5Hunan Key Laboratory of Animal Models for Human Diseases, School of Life Sciences, Central South University, Changsha 410013, China; 6Hunan Key Laboratory of Medical Genetics, School of Life Sciences, Central South University, Changsha 410013, China

**Keywords:** frogs, peptides, skin wounds, HSF cells, HUVEC cells

## Abstract

Amphibian skin contains wound-healing peptides, antimicrobial peptides, and insulin-releasing peptides, which give their skin a strong regeneration ability to adapt to a complex and harsh living environment. In the current research, a novel wound-healing promoting peptide, PM-7, was identified from the skin secretions of *Polypedates megacephalus*, which has an amino acid sequence of FLNWRRILFLKVVR and shares no structural similarity with any peptides described before. It displays the activity of promoting wound healing in mice. Moreover, PM-7 exhibits the function of enhancing proliferation and migration in HUVEC and HSF cells by affecting the MAPK signaling pathway. Considering its favorable traits as a novel peptide that significantly promotes wound healing, PM-7 can be a potential candidate in the development of novel wound-repairing drugs.

## 1. Introduction

Skin, as a physical barrier, protects internal organs and tissues from external harm such as mechanical or physical damage, as well as pathogenic micro-organisms [1,2]. Skin includes the keratinized lamellar epidermis and the underlying dermal connective tissue, which is collagen-rich and able to provide support and nourishment [3,4]. Skin is vulnerable to accidental damage [5], metabolic dysfunction [6], and skin diseases [7]. Wound healing after injuries is important for human health and survival. Four stages are included in wound healing: hemorrhage, inflammation, proliferation, and tissue remodeling [8,9,10]; each of which play a vital role in ensuring the restoration of skin integrity and functional reconstruction [11]. Small-molecule chemicals from plants, and proteins represented by epidermal growth factors are the two main groups involved in wound healing [12,13,14,15,16]. Considering the unstable activity and high cost of these drugs, it is critical to develop new drugs with the function of wound healing.

Among vertebrates, amphibians live in a complicated environment. Their skins are more challenged by biotic or abiotic factors [17,18]. During physiological and pathological processes, amphibian skin, limbs, and tails can heal rapidly and show a strong regenerative ability [17,19]. In addition, it has been reported that fresh frog skin can effectively promote wound healing [19]. Bioactive compounds derived from amphibian skin have been used in traditional and folk medicine for hundreds of years [20]. Amphibians may be a rich resource pool of bioactive compounds and, up to now, more than 100 peptides from the skin of 2000 species have been reported, including antimicrobial peptides [21,22], neuropeptides [23,24,25], wound-healing peptides [26,27,28], and insulin-releasing peptides [29,30].

Herein, we reported a novel peptide PM-7 with an amino acid sequence of FLNWRRILFLKVVR from the frog *Polypedates megacephalus*, which shows the activity of promoting wound healing in mice. Further, PM-7 can enhance cell proliferation and migration on Human Umbilical Vein Endothelial Cells (HUVEC) and Human skin fibroblast (HSF) cells and affect the MAPK signaling pathway, which exhibits an important function as an effective wound healing regulator.

## 2. Results

### 2.1. Purification and Identification of PM-7

The skin secretion of the frog *Polypedates megacephalus* was collected by anhydrous ether stimulation. The skin secretion was divided into four fractions using gel filtration (Figure 1A), and those fractions showing increased cell proliferation were purified with RP-HPLC C18 column (Figure 1B,C). The sample that displayed the activity of promoting cell proliferation was determined with a molecular weight of 1860.32 Da via a MALDI-TOF MS analysis (Figure 1D) and named PM-7. Using Edman degradation, PM-7 was identified as a new peptide containing 14 residues, with a sequence of FLNWRRILFLKVVR, which is consistent with the molecular mass.

### 2.2. PM-7 Accelerated Cell Proliferation and Migration in HSF and HUVEC Cells

During wound healing, the proliferation and movement of fibroblasts can greatly promote the process of re-epithelialization and wound healing. Our results showed that PM-7 significantly enhances the proliferation of HSF cells and the effect displayed a significant correlation with the peptide concentration. The proliferation rates of HSF cells increased by 150% and 180% at 500 μg/mL and 1000 μg/mL of PM-7, respectively, but PM-7 at 2000 μg/mL reduced its activity of promoting cell proliferation compared to that at 1000 μg/mL (Figure 2A). For HUVEC cells, PM-7 at 500 μg/mL slightly increased the rate of cell proliferation and at 1000 μg/mL dramatically enhanced the cell proliferation by 80%. Similar to HSF cells, PM-7 at 2000 μg/mL displayed a moderate effect on the proliferation of HUVEC cells (Figure 2B). We also explored the effect of PM-7 on cell migration using a cell scratch assay in vitro. As shown in Figure 2C, after 16 h of treatment, PM-7 enhanced cells migration across the sound chasm compared with the control treatment. Therefore, PM-7 accelerated cell proliferation and migration of HSF and HUVEC cells.

### 2.3. Effect of PM-7 on MAPK Signaling Pathway

The MAPK signaling pathway has been proven to be involved in promoting wound healing. We next carried out Western blot analyses to further investigate whether PM-7 could affect the MAPK signaling pathway in HUVEC cells in vitro or not. As shown in Figure 3, PM-7 significantly increased ERK and p38 phosphorylation after 12 h of 1000 μg/mL treatment. The results of Western blotting showed that the addition of PM-7 increased the expression of p-ERK1/2 and p-P38 that are the membranes of MAPK family.

### 2.4. PM-7 Enhanced the Healing of Full-Thickness Wounds in Mice

Since PM-7 showed activity in terms of promoting cell proliferation on HSF and HUVEC cells in vitro, the effect of PM-7 in vivo on the healing of full-thickness skin wounds was investigated by using an excision wound healing test in mice. The effect of PM-7 on full-thickness skin wounds in mice is shown in the figure. The application of PM-7 significantly accelerated wound closure in mice compared with that of treatment by PBS (Figure 4). On post-injury day 3, the wound area of mice treated with PM-7 was 50% smaller than that of the PBS-treated mice (Figure 4), suggesting that PM-7 accelerated wound healing. On post-injury day 5, the wounds treated with PM-7 and epidermal growth factor (EGF) were almost closed, while the wounds of control mice remained 45% open (Figure 4). After seven days, there was no obvious wound in PM-7-treated mice; however, the wounds in PBS-treated mice remained visible for several days (Figure 4). More importantly, there were no adverse effects on the body weight, overall health status, or behavior of the mice during the treatment with PM-7.

## 3. Discussion

There are plentiful bioactive peptides in the skin secretions of amphibians, including antimicrobial peptides, wound-healing peptides, lectins, and protease inhibitors, as reported before [25,27,31,32]. It has also been reported that the skin secretions of amphibians could accelerate wound healing because the skin often suffers various injuries, and some peptides have been identified to be beneficial to wound healing, such as a short peptide, CW49, from the skin of the frog *Odorrana graham* [26]. Here, we discovered a novel peptide PM-7 from the skin secretions of *Polypedates megacephalus.* For the isolation of skin secretions, gel filtration chromatography was carried out first to separate the different fractions via molecular weight of the different fractions. Specifically, some peptides or small-molecule chemicals came out later, as fraction IV (Figure 1A). Following that, reverse-phase high-performance liquid chromatography (RP-HPLC) was performed to further purify the peptide PM-7 (Figure 1B,C). Through a sequence alignment, we found that PM-7 does not share a similar structure to any previously discovered peptide, suggesting that PM-7 may be a novel candidate for a wound-healing drug.

Keratinocytes and fibroblasts are circuits that repair the dominant cells involved in the proliferation phase of wound healing [17,33,34]. Here, we found that PM-7 could accelerate the proliferation of HSF cells and HUVEC cells in a dose-dependent manner (Figure 2). Notably, the acceleration of cell proliferation was delayed at a high concentration of PM-7, which may be due to the complicated mechanisms of wound healing. The MAPK signaling pathway is sensitized by different intracellular- and extracellular-related substances, including peptide growth factors, cytokines, and related hormones, and can regulate various cellular activities, including proliferation, differentiation, survival, and death [25,35]. We checked the effect of PM-7 on the signaling pathway and found that PM-7 enhanced ERK and p38 phosphorylation in HUVEC cells (Figure 3). According to previous reports, some other peptides from amphibian skin also exert significant effects on wound healing through the MAPK signaling pathway [19,35].

Some peptides have been found from the skin of frogs, such as Cathelicidin-NV [19] and OM-LV20 [36]. PM-7 had a shorter sequence than some other peptides which display wound healing activity, which indicates that PM-7 would cost less if obtained with solid state synthesis. Importantly, PM-7 clearly enhanced the healing of full-thickness dermal wounds in vivo (Figure 4)*,* suggesting its possibility in clinical applications. Some growth factors used clinically, such as EGF, have been shown to enhance wound healing in a variety of tissues. The small peptide in this study, containing only 14 amino acid residues, might be a potential biomaterial or template for developing novel wound-healing agents.

In conclusion, PM-7 purified from the skin of *Polypedates megacephalus* is a bioactive/effector compound with potential wound-healing ability through the MAPK signaling pathway. It may facilitate the understanding of frog wound healing and skin regeneration. In addition, these properties make PM-7 a potent candidate for skin wound therapeutics.

## 4. Materials and Methods

### 4.1. Collection of Frog Skin Secretions and Isolation of Peptides

The frog skins were stimulated and purified as before [19], and the skin secretions were subjected to lyophilization and stored at −80 °C until use. Briefly, gel chromatography was carried out on a Sephadex G-50 gel filtration column (1.5 × 31 cm, superfine, GE Healthcare, Danderyd, Sweden) using 25 mM Tris-HCl buffer (pH 7.8) that contained 0.1 m NaCl. After pre-equilibrating the column, the skin secretions (500 μL, OD280 = 700 mAu), dissolved in a 25 mM Tris-HCl buffer, were eluted using the same buffer at a flow rate of 0.1 mL/min. A fraction collector (BSA-30A, HuXi Company, Shanghai, China) was used to automatically collect the fractions every 10 min and the fraction was checked at 280 nm. A reversed-phase high performance liquid chromatography (RP-HPLC) C18 column (Hypersil BDS C18, 4.0 × 300 mm, Elite, Shanghai, China) was pre-performed with water containing 0.1% (*v*/*v*) trifluoroacetic acid (TFA), followed by eluting along a linear gradient (0–70% acetonitrile, ACN in 70 min) with 0.1% (*v*/*v*) TFA and monitored at 280 nm. A mass spectrometer (MS) (Autoflex speed TOF/TOF, Bruker Daltonik GmbH, Leipzig, Germany) was used to analyze the purification of the crystallized samples and identify the molecular weight of the peptide. After lyophilization, the peptide was stored at −20 °C until use.

### 4.2. Edman Degradation Sequencing

Edman degradation was performed to determine the sequence of the peptide PM-7 using a PPSQ-31A protein sequencer (Shimadzu, Kyoto, Japan) according to the manufacturer’s standard protocols for GFD.

### 4.3. Cell Culture

HSF and HUVEC cells were cultured in Dulbecco’s Modified Eagle Medium and Ham’s F12 Medium (DMEM/F12) (BI, Beit Haemek, Israel) with 10% (*v*/*v*) fetal bovine serum (FBS, Hyclone, Logan, UT, USA) and antibiotics (100 units/mL penicillin and 100 units/mL streptomycin) in an incubator of 5% CO_2_ at 37 °C.

### 4.4. Cell-Scratch Healing, Migration, and Proliferation

The cell-scratch healing and proliferation of HSF and HUVEC cells were carried out to assess the cellular pro-healing activities of PM-7. Briefly, HSF cells were cultured overnight in a 24-well plate at a density of 2 × 10^6^ cells/well. Sterile 200-μL pipette tips were used to scratch the cell monolayer. Cells were washed with phosphate-buffered saline (PBS) solution three times before being cultured using a medium with PM-7 in different concentrations (0, 50, 100, 250, 500, 1000, and 2000 μg/mL). An inverted microscope (Motic, AE2000, Shenzhen, China) was used to obtain the images at different times. The methods for exploring the effect of PM-7 on the proliferation of HSF and HUVEC cells were the same as above. Briefly, 96-well plates were used to culture HSF and HUVEC cells at a density of 4 × 10^4^ cells/well, and they were exposed to PM-7 at different concentrations for 24 h. An MTS cell proliferation assay kit from Promega (Madison, WI, USA) was used to assess the impact of PM-7 on the proliferation.

### 4.5. Full-Thickness Skin Wounds in Mice

Full-thickness skin wounds in mice were created to check the regenerative effects of PM-7 in vivo. Male Kunming mice were obtained from Hunan Slack Jingda Experimental Laboratory Animal Co., Ltd. (Chansha, Hunan, China). The mice were given an adaptive feeding with standard food and water for one week and anesthetized through the peritoneum using 1% pentobarbital sodium (0.1 mL/20 g body weight). The corresponding hairs were removed from the back surface and a full-thickness wound (8 mm in diameter) was made on the dorsal skin of the back of every mouse. PBS, PM-7 (20 μL, 500 μg/mL) or EGF (20 μL, 100 μg/mL) was applied directly to the wound site on mice twice a day from day 0 to day 8. A digital photograph was used to record the wound condition every two days. ImageJ software v1.51 was used to calculate the wound area (percentage of residual wound area to primitive area) and quantified using GraphPad Prism v.8 (GraphPad Software Inc., San Diego, CA, USA). All animal experiments were approved and implemented in accordance with the requirements of Central South University.

### 4.6. MAPKs Signaling Pathway

HUVEC cells (2 × 10^6^) were cultured in six-well culture plates for 4 h. The adherent cells were seeded in a serum-free medium with the treatment of PM-7 for 12 h. After treatment with lysate (radio immunoprecipitation assay (RIPA)) containing phenylmethylsulphonyl fluoride (PMSF, Meilun Biotechnology, Dalian, China) and phosphatase inhibitor (Roche, Shanghai, China), cells were quantified using a BCA protein analysis kit (Meilun, Dalian, China). After separation of the cell samples using 12% SDS-PAGE, Western blotting was performed to analyze the protein samples and electroablation onto polyvinylidene fluoride (PVDF) membranes. ImageJ was used to check the grayscale and quantify the protein expression and degree of phosphorylation.

### 4.7. Statistical Analysis

Data are expressed as means ± S.E.M. The animals were randomly assigned (1:1:1 allocation) to receive PBS, EGF, or PM-7. For the cellular and animal experiments, two-tailed paired t-tests were used to compare two groups, and one-way ANOVA followed by Tukey’s post hoc test was used to evaluate differences among more than two groups. Statistical analyses were performed using GraphPad Prism v.8 software (GraphPad Software Inc., San Diego, CA, USA). A value of *p <* 0.05 was considered to be statistically significant.

## Figures and Tables

**Figure 1 toxins-14-00753-f001:**
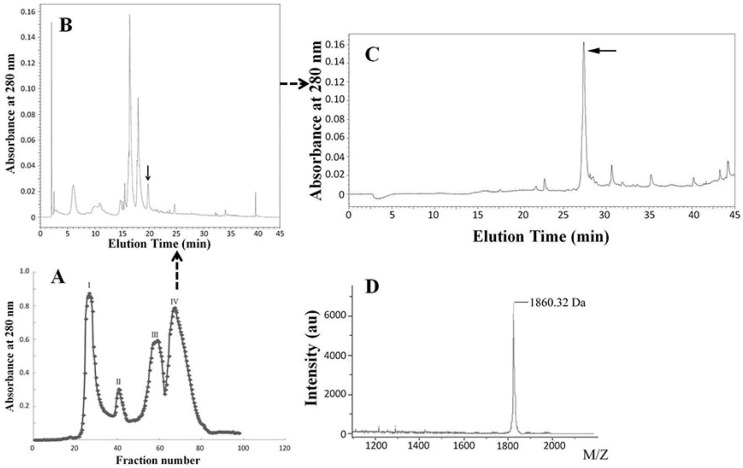
Isolation and purification of PM-7. (**A**) The skin secretion was isolated using G50 gel filtration and monitored at 280 nm, and the fraction IV displaying the activity of promoting cell proliferation is indicated with the arrow. (**B**,**C**) PM-7 was purified with RP-HPLC C18 and monitored at 280 nm. (**D**) The molecular weight of purified PM-7 was determined by mass spectrometry (1860.32 Da).

**Figure 2 toxins-14-00753-f002:**
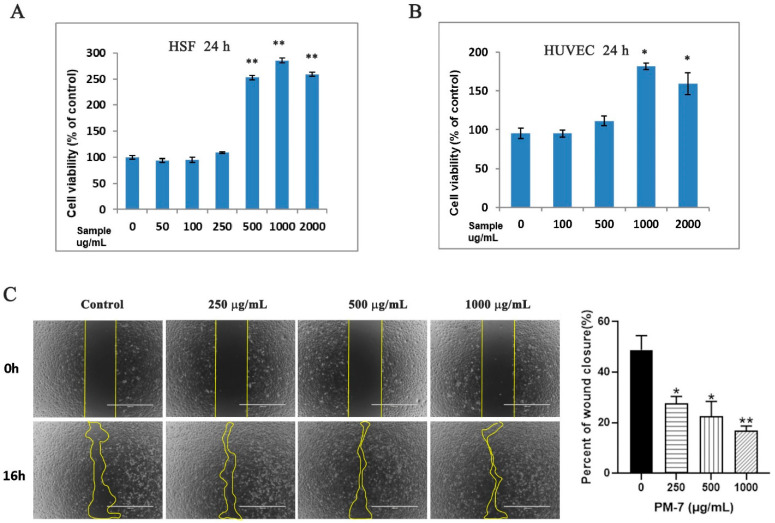
The effect of PM-7 on HSF and HUVEC cells. (**A**,**B**) Concentration-dependent effects of PM-7 on cell proliferation in HSF and HUVEC cells. (**C**) Representative images of PM-7 promoting healing of cell scratch on HSF cells. The line in each image represents the magnitude of 200 μm. Data represent the average of three independent experiments, expressed as a mean ± S.E.M., *n* = 3, ns: no statistical significance. * *p* < 0.05, ** *p* < 0.01. There were significant differences compared to the control group.

**Figure 3 toxins-14-00753-f003:**
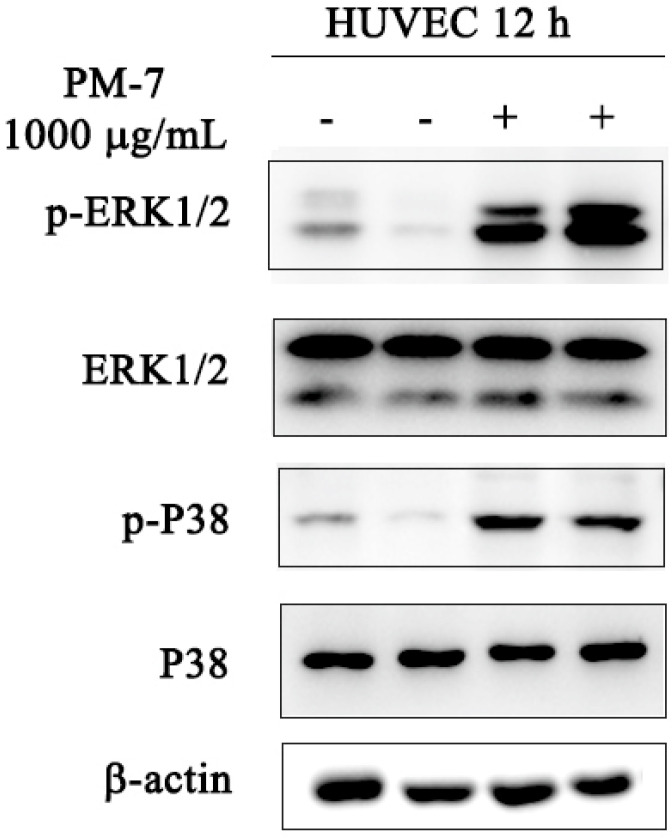
Effect of PM-7 on the MAPK signaling pathway in HUVEC cells using Western blot analyses. After a 12-h treatment with PM-7, the protein levels of p-ERK1/2 and p-P38 and the total ERK1/2 and p38 in HUVEC cells were detected using Western blot. β-actin was used to determine the amount of loaded protein.

**Figure 4 toxins-14-00753-f004:**
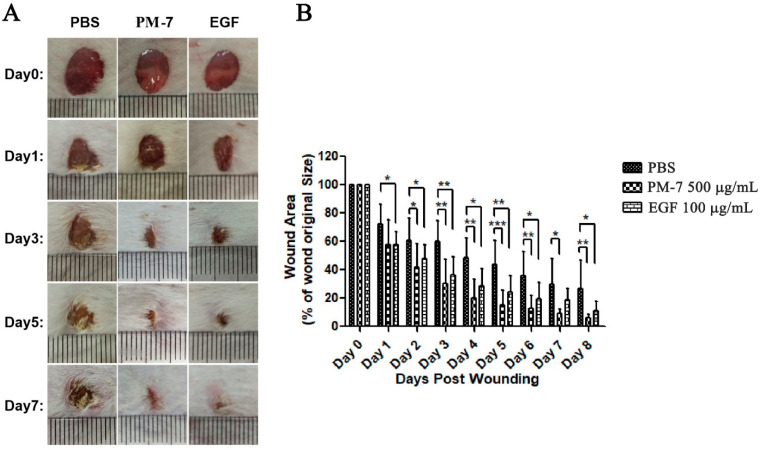
The effect of PM-7 on full-thickness skin wounds in mice. (**A**) Representative images of skin wounds on days 0, 1, 3, 5, and 7 with PM-7 treatment. PBS was a negative control and EGF was a positive control. A small cell on the ruler represents 1 mm. (**B**) Quantitative data showing the effect of PBS, PM-7, and EGF on full-thickness skin wounds in mice. Data are expressed as mean ± S.E.M., *n* = 6. * *p* < 0.05, ** *p* < 0.01, *** *p* < 0.0001.

## Data Availability

All data generated or analyzed during this study are included in this manuscript.

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
