# Peer review of "A Novel Peptide from Polypedates megacephalus Promotes Wound Healing in Mice"

_toxins, 2022, doi:10.3390/toxins14110753_

Round 1

Reviewer 1 Report

This manuscript describes the isolation of a novel peptide with wound-healing properties. The paper is reasonably well written, but could use some editing from a native English speaker. There are a lot of words that are not used correctly or misspelled (e.g. HUVEC). In addition, there are some issues that need to be addressed:

1) In the introduction, the authors should provide some context for other wound-healing peptides that have been discovered and are reported in the literature. For example, reference should be made to recent reviews or other papers, e.g. Acta Biomater. 103:52-67 (2020).

2) The peptide is not identified in Figure 1C. Which peak is it?

3) Figure 2: The experiment should be conducted with a positive control, so that one can compare the results obtained with the peptide versus some other known wound-healing agent.

4) Finally, the discussion should contain a short segment where the results are compared to other antimicrobial peptides or wound-healing agents. 

Reviewer 2 Report

This manuscript describes the isolation of a new peptide from the frog Polypedates megacephalus that promotes wound healing, both in vitro and in vivo in a mice model. The topic is important with a good potential for developing it for clinical use; however, the manuscript is scientifically poorly written; thus making it almost impossible to follow and evaluate.

- Firstly, a very extensive English editing is required!

- It is completely not clear how the PM-7 peptide was purified. How the frog skins were stimulated (line 45; as we did before??? Insert a reference or describe it shortly. Following lyophilisation; in which buffer was the dry material dissolved?? Which kind is C18 RP-HPLC column? What is the final solution/solvent of the peptide? Please provide a full protocol for its purification.

- Cell scratch healing, migration and proliferation; what was the control for the test samples? What is 0 sample? The correct control should be cells + the solvent of the PM-7 peptide at the same volume. A positive control for cell proliferation, such as another known wound healing reagent, should also be included in the assays; to have a sense of efficacy of the new peptide.

- How were the mice treated twice a day? Locally-how; were the peptide and EGF given in solution? at what volume? Please state clearly the treatment protocol.

- The authors should be careful when comparing the healing effect of the peptide and EGF in mice; as each reagent was tested only at one concentration and from the reported experiment here; it seems that both are showing very similar results.

- It is suggested to exchange sub-titles 3.3 and 3.4; the in vitro effect of PM-7 peptide should be reported first and the in vivo experiment the last one in the “Results” section.

- The discussion should be less a repeat of the results and more discussing the application possibilities of PM-7; other than wound healing? Any advantages of PM-7 as compared to other known peptides and etc.       

Minor corrections

- Figure 2: The line in each image of Figure 2C-what does it represent? Magnitude? Please add to the legend. Please label the three graphs/images with A, B, C.

- Figure 3: left graph-500ul/ml of what? Please add PM-7.

Reviewer 3 Report

The authors present here an interesting paper regarding a novel peptide from Polypedates megacephalus which promotes wound healing in mice. The MS is very interesting, but careless prepared. Nevertheless, the Reviewer feels it can be accepted after major amendments.

Specific comments:

Line 45: You should add a reference

Line 87: This sentence is confusing (...PBS is a control, not drug)

Line 94: It is interesting to find out only in the results section that the authors used 1000 nM concentration of PM-7. Please add this information here. Why are µg/ml given throughout the paper, but here just nM ?

Lines 122-127: µg/ml instead of ug/ml

Figure 2: There is no statistical information on the graphs. Can the authors make engravings with better resolution (for example using GraphPad as Figure 3), correct signing of X and Y axis (µg/ml), and marking the graphs as A-C ? In addition, the scale on the figure is missing. I suggest to the authors that they add a representative graph to Figure C, showing the area in which migrating cells were not observed. That way, differences can be shown and, most importantly, whether they are statistically significant or not.

Lines 149-151: Please, rewrite the sentence.

Line 152: In my opinion, no such conclusion can be drawn here. The authors used two different concentrations (PM-7 = 500 µg/ml; EGF = 100 µg/ml). If the concentrations were the same, then I would agree and the conclusion is correct. Here, unfortunately, it is not.

Figure 3: Please improve the resolution of the chart. In addition, the scale on the figures is missing.

A poorly written discussion. It practically repeats itself with what is written in the results. I suggest improving it. Very interesting is the information about the impact of PM-7 on the MAPK signalling pathway. Nevertheless, the authors did not make a minimal effort to clarify this issue.

In addition, the final conclusions of the paper are missing.

Reviewer 4 Report

The manuscript entitled: “A novel peptide from Polypedates megacephalus promotes wound healing in mice” is an interesting work for it shows a small peptide, called PM-7, with properties that can potentially make this peptide to be a wound healing drug candidate, as the author states. The results in section 3.3 show that PM-7 “ significantly accelerated the wounds in mice” (line142). The activity of PM-7 is also showed through the other experiments presented in this work. They seem to be well designed and controls properly presented.

However, as it is, I would not recommend the manuscript for publication. There are few points that need to be improved and/or explained:

- First of all, it is not just a suggestion, but it is absolutely necessary to have a thorough grammatical/language review of the full text by a native English speaker. Starting from the Abstract down to Discussion, the text is full of typos and grammatically incorrect sentences. I will not show all of them, since they are everywhere but as examples:

            - Lines 25-27: “Some small molecule chemicals from plants or proteins typified by epidermal growth factors are two main groups for wound healing” what does this mean? What are these two main groups? Small molecules form plants or proteins?

            - Line 29: “…amphibians live in more complex environments…” what are these complex environments?

            - Line 34: “ Amphibians may be a rich resource pool…” of bioactive compounds as suggested in previous paragraph?

            - Lines 49-50: “An automatic fraction collector (BSA-30A, HuXi Company, Shanghai, China) was carried out to collect the fractions every 10 min and the fraction was checked at 280 nm.” Was the collector carried out? Out of the lab? Did the collector collected only one fraction at 10 minutes?

            - Line 84: “ The mice were performed an adaptive feeding for one week…”

            - Line 92: “ All animal procedures were approved and conducted in accordance with the requirements of Centra South University.” Should it be Central South?

            - Line 104: “ 3.Result” Only one result?

And again, the whole text is full of this kind of language issues that need to be addressed.

Aside of this there are other questions for which an explanation is suggested:

- Line 45: “… as we did before…” It needs reference

- Line 88: “The different drugs (PBS, PM-7 and EGF) was used to treat the wounds twice a day from days”. Aside for the number of the verb (drugs was…), the authors consider PBS as a drug though they describe it earlier in line 73 as phosphate-buffered saline, which is its usual definition but it is not a drug. And, ok, we can accept EGF (epidermal growth factor) as a drug.

- The peptide used for this work is called PM-7. The authors mention a previous work, which I have not seen but, if they have a novel peptide and they called PM-7, what happened with PM-1, PM-2, etc.? Are they also active peptides, inactive? Are they not novel?  Do they look very different to PM-7? In lines 182-183 the authors mention PM-7 does not share “the” structure with any peptides before so we can guess that certainly peptides PM-1 to PM-6 are very different. Again, we can look it up but I would suggest a brief explanation of the other PM compounds (activity and structure for instance).

Round 2

Reviewer 1 Report

The authors have addressed the reviewer's concerns adequately.

Author Response

Thank you for the review.

Reviewer 2 Report

One more techanical correction is needed:  The authors  indeed exchanged sub-titles 3.3 and 3.4, as suggested; the in vitro effect of PM-7 peptide is reported first and the in vivo experiment the last one in the “Results” section. However, they didn't changed Figures 3 and 4. Old Fig. 4 should be now Fig. 3 and old Fig. 3 is now Fig. 4.

Author Response

One more techanical correction is needed:  The authors indeed exchanged sub-titles 3.3 and 3.4, as suggested; the in vitro effect of PM-7 peptide is reported first and the in vivo experiment the last one in the “Results” section. However, they didn't changed Figures 3 and 4. Old Fig. 4 should be now Fig. 3 and old Fig. 3 is now Fig. 4.

Response: Thanks for the suggestion. We have revised it.

Reviewer 3 Report

The Authors have tried to clarify most of the questions raised in the first review process. However, the article needs further revision.

The Materials and methods section lacks information on the statistical analyses conducted. So I suggest adding a Statistical Analysis section.

µg/ml instead of ug/ml - e.g. in Figure 2. Please check your entire article.

Figure 2 is still poorly presented. The authors indicated that they had made corrections, but the figure had not been corrected. 

I suggested to the authors that they add a representative graph to Figure C, showing the area in which migrating cells were not observed. That way, differences can be shown and, most importantly, whether they are statistically significant or not. The authors indicated that they had made corrections, but the figure had not been added.

Inserting literature footnotes in the text does not comply with the requirements of the journal. References are also incorrectly prepared. Please correct it.

And other errors as in line 96 (2 × 106). 6 should be written with a superscript.

For future reference, I suggest authors use the option to track changes in the text.

Please, read your article one more time, slowly and carefully.

Author Response

The Authors have tried to clarify most of the questions raised in the first review process. However, the article needs further revision.

The Materials and methods section lacks information on the statistical analyses conducted. So I suggest adding a Statistical Analysis section.

µg/ml instead of ug/ml - e.g. in Figure 2. Please check your entire article.

Response: Thanks for the suggestion. We have revised it.

Figure 2 is still poorly presented. The authors indicated that they had made corrections, but the figure had not been corrected.

I suggested to the authors that they add a representative graph to Figure C, showing the area in which migrating cells were not observed. That way, differences can be shown and, most importantly, whether they are statistically significant or not. The authors indicated that they had made corrections, but the figure had not been added.

Response: Thanks for the suggestion. We have revised it.

Inserting literature footnotes in the text does not comply with the requirements of the journal. References are also incorrectly prepared. Please correct it.

And other errors as in line 96 (2 × 106). 6 should be written with a superscript.

For future reference, I suggest authors use the option to track changes in the text.

Response: Thanks for the suggestion. We have revised it.

Reviewer 4 Report

It seems the authors of the manuscript entitled: A novel peptide from Polypedates megacephalus promotes wound healing in mice”, have addressed thoroughly the suggested corrections and clarifications. In this regard, the manuscript is acceptable for publication.

Author Response

Thank you for the review.